# Optimization of *Rhodococcus erythropolis* JCM3201^T^ Nutrient Media to Improve Biomass, Lipid, and Carotenoid Yield Using Response Surface Methodology

**DOI:** 10.3390/microorganisms11092147

**Published:** 2023-08-24

**Authors:** Selina Engelhart-Straub, Martina Haack, Dania Awad, Thomas Brueck, Norbert Mehlmer

**Affiliations:** Werner Siemens-Chair of Synthetic Biotechnology, Department of Chemistry, TUM School of Natural Sciences, Technical University of Munich, 85748 Garching, Germany

**Keywords:** *Rhodococcus*, response surface methodology, central composite design, media optimization, FAMEs, lipids, carotenoids

## Abstract

The oleaginous bacterium *Rhodococcus erythropolis* JCM3201^T^ offers various unique enzyme capabilities, and it is a potential producer of industrially relevant compounds, such as triacylglycerol and carotenoids. To develop this strain into an efficient production platform, the characterization of the strain’s nutritional requirement is necessary. In this work, we investigate its substrate adaptability. Therefore, the strain was cultivated using nine nitrogen and eight carbon sources at a carbon (16 g L^−1^) and nitrogen (0.16 g L^−1^) weight ratio of 100:1. The highest biomass accumulation (3.1 ± 0.14 g L^−1^) was achieved using glucose and ammonium acetate. The highest lipid yield (156.7 ± 23.0 mg g^−1^_DCW_) was achieved using glucose and yeast extract after 192 h. In order to enhance the dependent variables: biomass, lipid and carotenoid accumulation after 192 h, for the first time, a central composite design was employed to determine optimal nitrogen and carbon concentrations. Nine different concentrations were tested. The center point was tested in five biological replicates, while all other concentrations were tested in duplicates. While the highest biomass (8.00 ± 0.27 g L^−1^) was reached at C:N of 18.87 (11 g L^−1^ carbon, 0.583 g L^−1^ nitrogen), the highest lipid yield (100.5 ± 4.3 mg g^−1^_DCW_) was determined using a medium with 11 g L^−1^ of carbon and only 0.017 g L^−1^ of nitrogen. The highest carotenoid yield (0.021 ± 0.001 Abs_454nm_ mg^−1^_DCW_) was achieved at a C:N of 12 (6 g L^−1^ carbon, 0.5 g L^−1^ nitrogen). The presented results provide new insights into the physiology of *R. erythropolis* under variable nutritional states, enabling the selection of an optimized media composition for the production of valuable oleochemicals or pigments, such as rare odd-chain fatty acids and monocyclic carotenoids.

## 1. Introduction

A number of bacteria, some of which belong to the actinomycetes group, such as *Streptomyces*, *Rhodococcus*, and *Mycobacterium*, can accumulate lipids and triacylglycerols (TAG) when grown under nitrogen limitation [1]. These single-cell oils (SCOs) have the potential to serve as sustainable alternatives to fossil fuels [2]. Some bacteria can also inherently produce carotenoids. These isoprenoids range in color from yellow to red. Due to their antioxidant nature, carotenoids, which are stored in the cell membrane, play a crucial role in photo quenching and protection of the cell from photodamage. Photo-oxidative damage is dangerous for vital cellular machinery, such as DNA, lipids, and proteins. Therefore, biotic and abiotic stress triggers many organisms to accumulate carotenoids [3,4]. Carotenoids are utilized in a wide range of commercial industries, including the pharmaceutical, food, feed, and cosmetics industries [5]. The Gram-positive and aerobic bacteria of the diverse genus *Rhodococcus* offer a wide range of catabolic diversity and unique enzymatic capabilities [6,7]. Furthermore, the members of this genus can typically withstand various stress conditions, including metal toxicity and desiccation [8]. Due to this adaptability, *Rhodococcus* is a potential producer of various products, such as biosurfactants, bioflocculants, carotenoids, triacylglycerols, and antimicrobial compounds [8]. A range of micro-organisms, including microalgae, yeast and fungi, are able to produce lipids and carotenoids in higher quantities [9,10], with each species having specific advantages and disadvantages as producers. Microalgae or cyanobacteria can use carbon dioxide as their carbon source and produce photoautotrophic lipids and carotenoids, but they also have disadvantages, such as low lipid productivity and long fermentation duration for microalgae [11], as well as low carotenoid content for cyanobacteria [12]. *Rhodococcus* strains are of high interest as robust producers, as they can be genetically modified and harbor a large set of enzyme genes. As one of the few bacterial genera, *Rhodococcus* is only able to produce monocyclic carotenoids [8]. Further, the fatty acid profile revealed the production of rare odd-chain fatty acids [13]. The heterogeneous taxon *Rhodococcus* can be grouped into seven distantly related species-groups, i.e., A–G. All *R. erythropolis* strains belong to the species group D, which further includes *R. qingshengii* and several unclassified strains [14]. To enable the industrial application of *R. erythropolis*, an efficient genome-reduction tool was recently developed to produce a genome-structure-stabilized host strain. This strain allows the removal of unfavorable genes/functions [15]. Interestingly, *R. erythropolis* was initially classified as a non-oleaginous species due to its low lipid content when grown on glucose. However, recent studies have reported that this strain can exhibit an oleaginous and robust phenotype when grown on glycerol [16,17]. The carotenoids 4-keto-γ-carotene and γ-carotene were produced by *R. erythropolis* AN12, and in other *R. erythropolis* strains additional carotenoids, such as β-carotene, lycopene, and phytoene, have been identified [18,19]. The scarce research demonstrating lipid accumulation in this bacterium, as well as the limited studies investigating its carotenoid production under different nutrient sources, underscore the pressing need for further characterization to comprehensively elucidate its biology and explore its potential for the production of vital compounds. In this work, a wide range of nitrogen and carbon sources are tested in parallel to derive direct conclusions on their influence on yield of biomass, lipids, and carotenoids. Further, changes in the fatty acid profile are analyzed, thereby allowing the identification of the most promising application.

Identifying the optimal fermentation conditions is crucial to developing cost-efficient processes, as the medium composition directly correlates to product yield and productivity. The carbon source is considered to be the most essential medium component. It serves as an energy source and, therefore, plays a vital role in regulating the growth and production of primary and secondary metabolites. However, the selection and concentration of the nitrogen source cannot be neglected [20]. Nitrogen is an essential nutrient used in the synthesis of many cell components and metabolites. In yeast, inorganic nitrogen sources are better suited to biomass accumulation, while organic nitrogen sources have been shown to improve lipid production [21,22]. The precise ratio of carbon to nitrogen in the cultivation medium is crucial to improve lipid accumulation, as it typically occurs when micro-organisms are grown in excess of carbon, while other nutrients, particularly nitrogen, are limited [23]. Since growth is restricted under limited conditions, TAG accumulation mainly occurs in the stationary phase, as TAG are efficient storage compounds [1]. The one-factor-at-a-time (OFAT) optimization approach, which is commonly used in microbial systems, is very time consuming, consisting of iterative removal, supplementation, and replacement experiments. It is also unable to consider interactions between variables [20,24]. Statistics-based response surface methodology (RSM), on the other hand, allows rapid parameter screening at a reduced cost. RSM, which is a mathematical modeling algorithm developed by Box and Wilson [25], can be used as an experimental strategy to identify optimum conditions in a multivariable system. It is an efficient optimization technique, as it reduces the cost and time required to identify optimal cultivation conditions by reducing the number of examined conditions in an experimental design. Furthermore, this method enables defining the standalone effects of independent variables on the process, as well as their interactions [26,27]. Central composite design (CCD) has previously been utilized for the media optimization of *Rhodococcus* in various contexts, such as diesel oil [28] and aflatoxin B1 degradation [26] by *R. erythropolis*, the production of biosurfactant by *Rhodococcus* spp. MTCC2574 [29], and odd-chain fatty acid (FA) production by *Rhodococcus* sp. YHY01 [13]. CCD has not previously been used to optimize the production of lipids and carotenoids in *R. erythropolis*.

In the presented data, media optimization to increase biomass, lipid, and carotenoid accumulation of *R. erythropolis* JCM3201^T^ was performed via a threefold method: sequential investigation of nine distinct nitrogen sources and eight selected carbon sources was followed by the optimization of the carbon-to-nitrogen ratio via a circumscribed and rotatable CCD. The results shed light on the growth characteristics and FA profile of *R. erythropolis*, with the aim being to facilitate future studies of its use as a secondary metabolite producer.

## 2. Materials and Methods

### 2.1. Bacterial Strain and Inoculum Preparation

*R. erythropolis* JCM3201^T^ (DSM No. 43066, German Collection of Micro-Organisms and Cell Cultures GmbH, Braunschweig, Germany) was maintained using Luria–Bertani (LB) agar plates (10 g L^−1^ of peptone, 5 g L^−1^ of yeast extract, 10 g L^−1^ of sodium chloride, and 14 g L^−1^ of agar). For inoculum, single colonies were cultured in 500-milliliter baffled shake flasks holding 100 mL of LB medium at 120 rpm (New Brunswick InnovaTM 44, Eppendorf, Hamburg, Germany) for 48 h. Afterward, the inoculum was transferred to prospective optimization media.

### 2.2. Cultivation Medium

All cultures were cultivated in 500-milliliter baffled shake flasks holding 100 mL of ‘457. Mineral Medium’ (Brunner, DSMZ, Braunschweig, Germany). As a cultivation medium, ‘457. Mineral Medium’ (Brunner, DSMZ)) [30] (pH 6.9) was used, which contained 2.44 g L^−1^ of Na_2_HPO_4_, 1.52 g L^−1^ of KH_2_PO_4_, 0.20 g L^−1^ of MgSO_4_ × 7 H_2_O, 0.05 g L^−1^ of CaCl_2_ × 2 H_2_O, and 10 mL of trace element solution SL-4. Trace element solution SL-4 contained 0.50 g L^−1^ of EDTA, 0.20 g L^−1^ of FeSO_4_ × 7 H_2_O, and 100 mL of trace element solution SL-6. Trace element solution SL-6 contained 0.10 g L^−1^ of ZnSO_4_ × 7 H_2_O, 0.03 g L^−1^ of MnCl_2_ × 4 H_2_O, 0.30 g L^−1^ of H_3_BO_3_, 0.20 g L^−1^ of CoCl_2_ × 6 H_2_O, 0.01 g L^−1^ of CuCl_2_ × 2 H_2_O, 0.02 g L^−1^ of NiCl_2_ × 6 H_2_O, and 0.03 g L^−1^ of Na_2_MoO_4_ × 2 H_2_O. Carbon and nitrogen sources were individually selected for each experiment.

### 2.3. Experimental Design

Media optimization was divided into three stages, with the first and second stages investigating various nitrogen and carbon sources using an OFAT strategy. Nitrogen sources encompassed ammonium chloride, diammonium hydrogen phosphate, ammonium sulfate, potassium nitrate, ammonium nitrate, yeast extract, tryptone/peptone, urea and ammonium acetate (Table 1). Based on the supplier’s information (Carl Roth, Karlsruhe, Germany), yeast extract and tryptone/peptone contained 10.8% (*w*/*w*) and 12.3% (*w*/*w*) nitrogen, respectively. To examine the effect of the nitrogen source on growth, a medium, which contained 40 g L^−1^ of glucose (16 g L^−1^ carbon) and 0.16 g L^−1^ of nitrogen, was used. Carbon sources encompassed glucose, galactose, fructose, lactose, sucrose, maltose, sorbitol, and glycerol (Table 1). For the screening of carbon sources, cultivation was carried out in media using the optimal nitrogen source, as observed in the previous stage of optimization, which was ammonium acetate. For each nitrogen and carbon source, biological triplicates were prepared. Each culture was inoculated to OD_600nm_ 0.2 and cultivated at 28 °C and 120 rpm for 192 h.

Following the RSM strategy, different carbon-to-nitrogen concentrations and ratios were assessed. Subsequently, the effect of varying carbon-to-nitrogen sources on the growth, as well as lipid and carotenoid production of *R. erythropolis*, was compared. A total of 9 different carbon-to-nitrogen ratios were tested using a CCD. Concentrations ranged from 3.93 to 18.07 g L^−1^ elemental carbon and 0.02 to 0.58 g L^−1^ elemental nitrogen, with glucose and ammonium acetate used the as carbon and nitrogen sources, respectively. Carbon-to nitrogen ratios ranged from 12:1 to 647:1. For the center point (11 g L^−1^ of carbon and 0.3 g L^−1^ of nitrogen), five biological replicates were prepared; all other ratios were prepared in biological duplicates. Each culture was analyzed in two technical replicates. Cultures were inoculated to OD_600_ 0.2 and then cultivated for 192 h (28 °C, 120 rpm).

### 2.4. Growth Analysis

Optical density was measured at 600 nm using a photometer (Nano Photometer NP80, IMPLEN, Munich, Germany). Standard semi-micro cuvettes made of polystyrene were used, and each cuvette had a path length of 10 mm.

For dry cell weight (DCW) analysis, 30 mL of culture was sampled after 140 and 192 h. For the CCD, 30 mL of culture was divided into two pre-weighed falcon tubes. Cultures were centrifuged (3500× *g*, 5 min), and the cells were washed using bidistilled water and lyophilized (−80 °C, ≥72 h). After gravimetric measurements, the weight of the empty vessels was subtracted from that of vessels containing lyophilized biomass.

### 2.5. Carotenoid Extraction

Carotenoid extraction from dry biomass was carried out as previously shown [31]. Depending on availability, between 2 and 10 mg biomass was weighed into 1.5-milliliter Eppendorf tubes. In brief, pigments were extracted using acetone after cell disruption via glass beads (2 mm). The absorbance at 454 nm was measured using a Nano Photometer NP80 (IMPLEN, Munich, Germany), and the titer calculated according to Formula (1).
Carotenoid titer (Abs_454nm_ mg^−1^_DCW_) = Absorbance (Abs_454nm_)/Dry Biomass (mg)(1)

### 2.6. Fatty Acid Profile

FA analysis was carried out according to the method described by Engelhart-Straub et al. [31]. In brief, lipids extracted from biomass were converted into fatty acid methyl ester (FAME) via MultiPurposeSampler MPS robotic (Gerstel, Linthicum Heights, MD, USA). FAMEs were measured via gas chromatography (GC-2025 coupled to an AOC-20i auto-injector and an AOC-20s auto-sampler, Shimadzu, Duisburg, Germany) and equipped with a flame ionization detector. For separation, a Zebron ZB-wax column (30 m × 0.32 mm, film thickness 0.25 μm, Phenomenex, Aschaffenburg, Germany) was used. The external standards, namely Marine Oil FAME mix (20 components from C14:0 to C24:1; Restek GmbH, Bad Homburg, Germany) and FAME #12 mix (C13:0, C15:0, C17:0, C19:0, and C21:0; Restek GmbH, Bad Homburg, Germany), were utilized for the identification and quantification of FAMEs. Normalization was based on the internal methyl laurate standard (C12; Restek GmbH, Bad Homburg, Germany).

For the identification of branched FA, GC–MS was performed using a Thermo Scientific™ TRACE™ Ultra Gas Chromatograph instrument coupled with a Thermo DSQ™ II mass spectrometer and a Triplus™ Autosampler injector. For this analysis, a BPX5 column (30 m × 0.25 mm, film thickness of 0.25 μm) acquired from von Trajan Scientific Australia Pty Ltd. (Ringwood, Victoria, Australia) was used. The separation was performed at an initial column temperature of 50 °C, which increased at a rate of 4 °C min^−1^ up to a final temperature of 250 °C. The used carrier gas was hydrogen at a constant flow rate of 0.8 mL min^−1^.

### 2.7. Response Surface Methodology and Further Statistical Analysis

Design-Expert Software, Version 22.0.2 (Stat-Ease, Inc., Minneapolis, MN, USA) was adopted for the design and analysis of the media optimization. A full central composite design with eight non-center points in duplicates and five center points, as well as an alpha value of 1.41421, was performed. The range of values is listed in Table 2. The analysis of variance (ANOVA) was used to estimate the appropriate statistical parameter and evaluate the suitability of the design. The effects of carbon and nitrogen levels on the level of the dependent variables—biomass (g L^−1^), lipid content (mg g^−1^_DCW_) and carotenoid titers (Abs_454nm_ mg^−1^_DCW_)—were analyzed. Non-significant terms not needed for hierarchy were excluded from the model.

## 3. Results and Discussion

### 3.1. Effect of Different Nitrogen Sources on Biomass, Lipid, and Carotenoid Formation of R. erythropolis

To determine the appropriate carbon and nitrogen concentrations and assess the effects of different carbon and nitrogen sources on *R. erythropolis*, a full central composite design-based DOE (design of experiment) analysis was performed. Glucose was selected as the carbon source, and ammonium sulfate was used as the nitrogen source. While increasing the carbon and nitrogen concentrations resulted in increased biomass formation, a low nitrogen concentration induced increased lipid production. To produce a significant amount of biomass while retaining nitrogen limitation at a later point in the cultivation, a C:N of 100:1, with 16 g L^−1^ of carbon and 0.16 g L^−1^ of nitrogen, was used for the following experiments, in which the effects of different nitrogen and carbon sources on *R. erythropolis* were examined.

This study examined nine distinct nitrogen sources, including ammonium chloride, diammonium hydrogen phosphate, ammonium sulfate, potassium nitrate, and ammonium nitrate, as defined inorganic sources. Yeast extract and tryptone/peptone were used as complex organic sources, in addition to urea and ammonium acetate, which were used as defined organic sources (Figure 1).

The cultivation proceeded for eight days under identical conditions, which allowed direct comparison. Yields of biomass, lipids, and carotenoids were determined after 140 and 192 h, as two timepoints enable a more detailed assessment of the differences between the cultures. Cultures containing any of the five defined inorganic sources exhibited slow growth at the beginning (Figure 1a). When inorganic nitrogen was used as nitrogen source, compared to complex sources, the micro-organism needed additional energy to perform amino acid synthesis [27]. The lowest biomass accumulation was achieved for potassium nitrate as the nitrogen source, with a DCW of 0.36 ± 0.08 g L^−1^ after 192 h. In contrast, a strain identified as *Rhodococcus* sp. BCH2 showed the highest growth on potassium nitrate, which was comparable to that of ammonium acetate [32]. The complex nitrogen sources’ yeast extract and tryptone/peptone showed the fasted growth from the start of the cultivation with no detectable lag phase. Complex nitrogen sources provide additional nutrients, including amino acids, polypeptides, vitamins, and trace elements. Increased yield of biomass with complex nitrogen sources was achieved for a variety of micro-organisms, including *C. oleaginosus* and *A. oryzae* [27,33]. When cultivating the *R. erythropolis* strains MTCC1548, 2794, and 3951 using different nitrogen sources, inorganic nitrogen sources (ammonium sulfate and ammonium phosphate), as well as urea, resulted in very low biomass compared to complex organic nitrogen sources (yeast extract and meat peptone) [29]. In this study, *R. erythropolis* cultures, which contained the two aforementioned complex nitrogen sources, entered the stationary phase after approximately 115 h (Appendix A). After 140 h, a DCW of 2.13 ± 0.07 g L^−1^ for yeast extract and 1.88 ± 0.03 g L^−1^ for tryptone/peptone was determined, with a decreased DCW after 192 h. The higher yield of the biomass when using yeast extract could be related to its high vitamin content relative to tryptone/peptone [34]. The final group of assessed nitrogen sources comprised defined organic sources in the form of urea and ammonium acetate. While urea induced an extended lag phase within the first 120 h, cultures with ammonium acetate exhibited a steady growth over 192 h and no lag phase. The highest biomass was achieved when ammonium acetate was used as the nitrogen source with a DCW of 2.84 ± 0.02 g L^−1^ after 192 h.

Since biomass productivity was the primary goal of these cultivations, an increased nitrogen content of 0.16 g L^−1^ was selected, leading to a lipid content of 5 to 16% in *R. erythropolis* after 192 h. The lipid content for all distinct nitrogen sources, except for potassium nitrate, increased between 140 and 192 h (Figure 1c). The highest lipid yield was determined for cultures grown using the complex nitrogen sources yeast extract and tryptone/peptone, which registered 156.7 ± 23.0 mg g^−1^_DCW_ and 132.3 ± 15.4 mg g^−1^_DCW_ after 192 h, respectively. As TAG accumulation primarily occurs in the stationary phase [1] and cultures grown on complex nitrogen sources entered the stationary phase after 115 h, a sufficient time period for lipid accumulation was allowed. In contrast, *R. erythropolis* exhibited a longer lag phase when grown using the five defined inorganic nitrogen sources, having a lipid content of 5–5.5% (g g^−1^_DCW_) after 192 h. Cultures grown on potassium nitrate showed a higher lipid content after 140 h, i.e., 10.5% (g g^−1^_DCW_). Inefficient utilization of this compound could potentially induce nitrogen limitation in these cultures during the early cultivation phase, which could explain the elevated lipid yield after 140 h. Cultures grown using ammonium acetate showed the highest DCW, having a lipid content of 68.3 ± 5.0 mg g^−1^_DCW_ after 192 h. In *R. toruloides*, organic nitrogen sources also led to increased lipid contents compared to inorganic nitrogen sources [35].

When *R. opacus* PD630 was grown using waste paper hydrolysate, the inorganic nitrogen sources ammonium chloride, followed by ammonium sulfate, gave the highest biomass and lipid yield, having a lipid content of around 25% after 120 h. When grown using the complex nitrogen sources urea, yeast extract, and peptone, biomass and lipid accumulation was lower. The lipid content of cultures grown using urea (15.24%) was reported to be three-fold higher than that of cultures grown using yeast extract (5.69%) and peptone (4.34%) [2]. These results are in contrast to the presented data in this study. One reason for this outcome could be the utilization of the complex carbon source waste paper hydrolysate, which offers a wide range of readily metabolizable ammonia acids, polypeptides, nucleotides, various vitamins, and trace elements. Therefore, the advantage of a complex nitrogen source, like yeast extract or tryptone/peptone, could be masked, as these substances are already available. Furthermore, *R. opacus* generally produces higher amounts of TAGS than *R. erythropolis* while cultivated using sugars, organic acids, and hydrocarbons [8].

The effect of different nitrogen and carbon sources on carotenoid accumulation in *R. erythropolis* has not previously been investigated. The carotenoid content of all cultures was assessed after 140 and 192 h (Figure 1d), with decreasing carotenoid contents recorded over this time period in all cultures. The highest carotenoid content was achieved when diammonium hydrogen phosphate (0.039 ± 0.006 Abs_454nm_mg^−1^_DCW_) was used as a nitrogen source, followed by ammonium sulfate. Cultures grown using ammonium chloride, potassium nitrate, ammonium nitrate, and ammonium acetate produced similar amounts of carotenoids after 140 h, having an absorbance of between 0.019 and 0.021 Abs_454nm_mg^−1^_DCW_. Temperature, pH, and carbon and nitrogen sources affect carotenoid production in micro-organisms [3,36,37]. *Rhodotorula glutinis* produced the highest amount of carotenoid using ammonium sulfate as nitrogen source, followed by yeast extract and peptone [34]. In contrast, for *Rhodotorula* sp. RY1801, the highest carotenoid production was achieved using yeast extract, followed by urea and ammonium sulfate. When ammonium nitrate was used as nitrogen source, the lowest carotenoid production was determined [36]. While the highest growth of the bacterium *Sphingobium* sp. was detected using diammonium hydrogen phosphate, complex nitrogen sources, like tryptone, soya peptone, and okara, were found to be the best options for the production of the carotenoid nostoxanthin [38]. The fungi *Umbelopsis ramanniana* produces, using diammonium hydrogen phosphate and the carbon source maltose, the highest amount of carotenoids compared to other nitrogen sources, while using the carbon source glucose, the lowest amount of carotenoids is produced [39]. Urea increased the production of total carotenoids and, in particular, β-carotene in *Flavobacterium multivorum*, likely by inhibiting hydroxylase activity, thereby decreasing the production of zeaxanthin [40]. While some studies indicated higher carotenoid yields using organic nitrogen sources, this finding could not be observed in *Microbacterium paraoxydans* or in the presented data in this study [38,41].

### 3.2. Effect of Different Carbon Sources on Biomass, Lipid, and Carotenoid Formation by R. erythropolis

For the evaluation of different carbon sources, ammonium acetate was selected as the nitrogen source due to the resulting highest achieved biomass accumulation and relatively high lipid yields. A defined nitrogen source offers the advantage of a controlled medium composition; therefore, possible fluctuations can be directly traced to the carbon source. Eight carbon sources were examined in this work, including glucose, galactose, and fructose, as monosaccharides/hexose. Lactose, sucrose, and maltose were included as disaccharides. Moreover, the sugar alcohol sorbitol and glycerol were examined (Figure 2). The latter element represents an interesting carbon source; as the main by-product of biodiesel production, it has a low cost and is readily available [16,42].

The cultivation proceeded for eight days under identical conditions of 16 g L^−1^ of carbon and 0.16 g L^−1^ of nitrogen in the form of ammonium acetate, resulting in a C:N of 100. No growth in *R. erythropolis* was detected for arabinose and xylose, which is congruent with other *Rhodococcus* strains. However, xylose, as well as arabinose utilization, was achieved via genetic engineering in *R. opacus* PD630 and *R. jostii* [43,44,45,46]. Xylose assimilation capabilities enable the efficient utilization of cellulose and hemicellulose degradation products [47], such as wheat bran hydrolysate [48] or corn stover hydrolysate [49]. Very low growth with final biomasses of under 0.5 g L^−1^ were determined for cultures with galactose, lactose, and maltose. The inability to grow on galactose and lactose was reported in other *Rhodococcus* strains, such as *R. jostii*, *R. erythropolis*, *R. fascians*, and *R. equi* [50]. The only strain capable of growth on these carbon sources was *R. opacus* [50], which is also reported to grow on maltose [51]. Further, *R. erythropolis* ZJB-09149 could grow using maltose, as well as lactose [52]. When fructose was used as a carbon source in this study, the fastest growth was measured for the first 48 h, after which point the stationary phase was entered. When grown using the two sugar alcohols sorbitol and glycerol, the strain entered the stationary phase two days later at around 94 h. For cultures grown using the disaccharide sucrose, the stationary phase also started after 94 h, with a final DCW of 2.47 ± 0.22 g L^−1^ noted after 192 h. The highest yield of biomass was achieved using glucose as a carbon source, with a DCW of 3.1 ± 0.14 g L^−1^ noted after 192 h. Interestingly, the stationary phase of cultures grown using glucose was not reached before 164 h, which was later than all other investigated carbon sources. When grown using different carbon sources, *R. erythropolis* strains MTCC1548, 2794, and 3951 showed distinct differences. While MTCC1548 grew best using glucose as carbon source, followed by glycerol, sucrose, and sorbitol, MTCC2794 and MTCC3951 showed the highest growth using sucrose, followed by sorbitol, glucose, and glycerol. Using glucose, MTCC2794 grew to around double the OD_600_ values of MTCC1548 and MTCC3951 [29]. When *R. erythropolis* LSSE8-1 was grown using different carbon sources, the highest OD_600_ was determined for glycerol, followed by sucrose and glucose, which is in reverse order to the growth investigated in this work [53]. The varying growth behaviors of *R. erythropolis* strains further emphasize the heterologous nature of these different strains.

The lowest lipid contents (≤3.6%) were measured for the cultures with galactose, lactose, and maltose, which also exhibited the lowest growth. Due to the low growth and, therefore, low consumption of nutrients, nitrogen limitation was presumably not reached. In comparison, fructose and glucose accumulated over double the amount of lipid. The highest lipid content was determined for the cultures grown using the sugar alcohols sorbitol and glycerol, which registered values of 93.8 ± 4.1 mg g^−1^_DCW_ and 86.1 ± 13.4 mg g^−1^_DCW_ after 192 h, respectively. Cortes et al. examined the effects of various carbon sources on the lipid accumulation in *R. erythropolis* DCL12 and *R. opacus* PWD14 cells. The experimental setup incorporated 0.01 g L^−1^ of nitrogen, and an excess of carbon (2 g L^−1^ of glucose) was used. *R. erythropolis* accumulated 78.71% (g g^−1^_DCW_) lipids in the stationary phase [54]. *R. erythropolis* IGTS8, when grown using 30 g L^−1^ of glycerol and 0.75 g L^−1^ of urea as carbon and nitrogen source, produced lipid content of 45.8% in 96 h [55].

Insufficient amounts of biomass were available to determine the carotenoid content of cultures grown using galactose and lactose. The highest carotenoid contents after 140 h were measured in cultures grown using maltose and glycerol, followed by fructose. The lowest carotenoid content was determined for cultures using glucose, with only around half of the carotenoids compared to the former. While *R. mucilaginosa* grew the fastest in the glucose medium, the highest carotenoid concentration was measured in the sucrose molasses-containing medium. The highest product yield was obtained using whey lactose as the carbon source [37]. This result is in agreement with those of *Phaffia rhodozyma* strains, which showed increased carotenoid production when using xylose, followed by sucrose, rather than glucose, as the carbon source [56]. In contrast, glucose was determined to be the most suitable carbon source for the production of carotenoids in *Rhodotorula* sp. RY1801, followed by fructose. Sucrose, lactose, and maltose showed comparable levels of carotenoid production [36]. *Umbelopsis ramanniana* produced the highest carotenoid amount with maltose used as the carbon source, followed by lactose and glucose [39].

In previous studies, it was demonstrated that *Rhodococci* can grow on various waste stream hydrolysates, ranging from sugar cane molasse, orange waste, and olive mill waste to cellulosic feedstocks [42], rendering them potent organisms for use in sustainable industrial processes. The data presented here regarding the bacterium’s behavior in a wide range of carbon and nitrogen sources offer a valuable resource for the identification of suitable and cost-effective growth media.

### 3.3. Effect of Nitrogen and Carbon Sources on the Fatty Acid Profile of R. erythropolis

The FA composition of *R. erythropolis* is heavily influenced by the cultivation conditions, including the carbon and nitrogen source, as well as the pH-value, temperature, and aeration [42]. FA represent an essential part of the phospholipid bilayer of the cellular membrane [57], which allows bacteria to adapt their lipid composition to maintain membrane fluidity and permeability in response to various stress conditions [58,59]. Bacteria of the actinomycetes group store large amounts of TAGs in lipid bodies. During lipid biosynthesis, an oleaginous layer is formed at the cytoplasm membrane, followed by the subsequent formation of lipid pre-bodies, which are finally released as mature lipid bodies in the cytoplasm [60].

The relative quantification of FAs is depicted as the percentage of total FAs (*w*/*w*) (Figure 3). Independent of the nitrogen or carbon source, the main components of the FA profile of *R. erythropolis* are palmitic acid (C16:0), 10-methyl octadecanoic acid (10-Me-18:0), myristic acid (C14:0), and oleic acid (C18:1). 10-Me-18:0 is a fully saturated and long-chain FA, which is characterized by a low melting temperature and high oxidative stability [61]. It occurs naturally in the membrane of a variety of actinobacteria [62]. In previous studies, 10-methyl octadecanoic acid was determined in *R. erythropolis* 3C-9 and DSM 43066 [63,64,65]. Tsitko et al. (1999) observed an increase of up to 34% of branched fatty acids (BFAs) of total FAs in *R. opacus* when cultivated using aromatic compounds. The role of 10-Me-18:0 in the protection of the membrane–cell wall structure against disruption was suggested [66].

The FA contents of all cultures grown using different nitrogen sources (Figure 1c), with the exception of cultures grown using potassium nitrate, increased between the samples taken after 6 (140 h) and 8 days (192 h). While the contents of palmitoleic acid (C16:1) decreased in all samples over time, the contents of palmitic acid (C16:0) increased in all samples, with exception of potassium nitrate, up to 34.1% in cultures grown using yeast extract and tryptone/peptone. Both nitrogen sources are associated with minimal differences between the two timepoints, i.e., 140 and 192 h. A distinct decrease in 10-methyl octadecanoic acid (10-Me-C18:0) was measured in samples grown using yeast extract (11.5%), tryptone/peptone (12.2%), and ammonium acetate (16.3%). In all other samples, contents of over 20% of this FA were determined. These three carbon sources also produced the highest contents of oleic acid (C18:1) and stearic acid (C18:0). Palmitic acid (C16:0) and oleic acid (C18:1) have been reported to be the main FAs in oleaginous *Rhodococci*, which produce high amounts of TAGs (50–75% g_lipid_ g^−1^_DCW_) [42].

The FA profile of cultures grown using distinct carbon sources were also compared (Figure 4). Distinct FA profiles were determined for cultures grown on lactose and maltose. Between 33 and 38% of the profile was identified as 10-methyl octadecanoic acid (10-Met-C18:0), more than double the amounts compared to the cultures using glucose, fructose or glycerol. Galactose, lactose, and maltose not only sustained reduced biomass, but also resulted in low levels of oleic acid (C18:1) (≤7.8%) and stearic acid (C18:0) (≤2.2%).

When comparing the FA profiles of cultures grown using glucose and glycerol as carbon sources, cultures cultivated using glycerol exhibited an increased amount of oleic acid (C18:1; 15.1 compared to 20.6%) and a decreased amount of 10-methyl octadecanoic acid (10-Me-C18:0; 15.7 compared to 11.9%). These findings are in line with those of Bhatia et al. (2019), who identified distinct FA profiles of three different *Rhodococcus* sp. species (YHY01, 1918, and 19938) grown using these carbon sources [13].

Odd-chain fatty acids (OCFA) are commercially important products due to their antifungal, anti-allergic, and anti-inflammatory properties [13,67,68]. Microbial oil typically comprises very low OCFA content [13,69], while *Rhodococcus* has the ability to accumulate high OCFA content, mainly pentadecanoic acid (C15:0) and heptadecanoic acid (C17:0). In *Rhodococcus* sp. YHY01, a maximum proportion of 85% *w*/*w* of all FA were identified as OCFA when cultivated using a mixture of glycerol, propionate, and ammonium chloride [13]. An increased amount of pentadecanoic acid (C15:0) could also be measured in cultures cultivated using sorbitol (2.4%) and glycerol (3.1%). Elevated amounts of heptadecenoic acid (C17:1) were determined in cultures cultivated using lactose (6.4%) and maltose (9.3%) compared to other carbon sources evaluated in this work.

### 3.4. RSM Model: Effect of Nitrogen and Carbon Concentrations on Biomass, Lipid and Carotenoid Formation of R. erythropolis

As a final step, the effects of a wide range of carbon and nitrogen concentrations on biomass accumulation, as well as lipid and carotenoids production, in *R. erythropolis* was assessed for the first time. To meet this goal, a full central composite design-based DOE (design of experiment) analysis was performed. For the CCD setup, glucose was selected as the carbon source, as the highest biomass was achieved with it (Figure 2). Good growth of *Rhodococcus* strains on glucose is well reported in the literature [70,71,72]. Ammonium acetate was used as the most promising nitrogen source. A C:N range of 12 to 160 was assessed, with a maximal C:N of 647 being part of the star points. The CCD design matrix, as well as the corresponding response on the dependent factors: biomass, lipid, and carotenoid contents, are illustrated in Table 3.

The fitted regression equations for the CCD model are represented below
Biomass-140 h (g L^−1^) = −0.145 + 0.076 C + 7.195 N(2)
Biomass-192 h (g L^−1^) = 1.228 − 0.059 C + 5.207 N + 0.588 C × N (3)
Lipid-140 h (mg g^−1^_DCW_) = 34.867 + 1.608 C − 99.460 N − 4.281 C × N + 192.195 N^2^(4)
Lipid-192 h (mg g^−1^_DCW_) = 76.816 + 0.946 C − 300.836 N + 343.229 N^2^(5)
Carotenoide-140 h (Abs_454nm_ mg^−1^_DCW_) = 0.019 − 0.0002 C − 0.016 N + 0.001 C × N(6)
Carotenoide-192 h (Abs_454nm_ mg^−1^_DCW_) = 0.015 − 0.0002 C + 0.024 N − 0.025 N^2^(7)

C and N represent carbon and nitrogen levels in g L^−1^, respectively. Table 4 showcases the ANOVA analysis of the reduced CCD models (insignificant factors excluded). The complete ANOVA analysis of the models after 192 h is listed in Appendix A. The regression equations after 192 h translate into the 3D response surface plots depicted in Figure 5.

It can be deduced from the formulated regression Equations (2) and (3) that both of the independent factors, i.e., carbon and nitrogen, greatly affect biomass accumulation. After 192 h, a 2FI model was found to most accurately explain the interactions by exclusively considering significant terms. Carbon (linear) and nitrogen (linear), as well as the interactions between both concentrations, significantly affect the yield of biomass. With increased concentrations of carbon and nitrogen, increased biomass accumulation was determined (Figure 5a). The highest biomass was achieved with 11 g L^−1^ of carbon and 0.58 g L^−1^ of nitrogen, which corresponds to the highest tested nitrogen concentration, with 8.0 ± 0.3 g L^−1^ being recorded after 192 h.

For the lipid content, a reduced quadratic model with carbon (linear) and nitrogen (linear and quadratic) concentrations resulted in the best fitted model after 192 h. An inverse trend regarding biomass accumulation was observed. With decreasing carbon and nitrogen concentration, increasing lipid formation was observed. Nitrogen limitation can lead to high lipid production in oleaginous micro-organisms, but it simultaneously limits the biomass formation, as can be especially observed for Run 13 (Table 3). This observation was shown for a range of micro-organisms, including the oleaginous yeast *Cutaneotrichosporon oleaginosus* [27] and *Rhodotorula glutinis* [73]. Star points (Run 9 to 15) represent extreme low, as well as high, values for each factor of the design. These points are typically applied to estimate the curvature of the model. Run 13 represents minimal nitrogen content in the medium. Here, the lowest biomass (0.5 ± 0.05 g L^−1^), as well as the highest lipid content (100.5 ± 4.3 mg g^−1^_DCW_ ), could be determined after 192 h. Due to a drastic decrease in the yield of biomass and increase in lipid formation in the culture compared to other tested media compositions, the model is unable to precisely predict this point, which could explain the significant lack of fit p-value for both lipid titer models (Table 4). Various C:N were tested for *R. opacus* PD630 grown using waste paper hydrolysate. While the biomass accumulation steadily decreased with an increase in C:N of 10 to 80, the lipid content steadily increased. A C:N of 60 yielded a lipid content of 41.6% [2].

The effects of carbon and nitrogen concentration on carotenoid production in *Rhodococcus* were investigated for the first time. When explaining the carotenoid content after 192 h, a reduced quadratic model with carbon (linear) and nitrogen (linear and quadratic) resulted in the best fit. Low carbon and high nitrogen content in the medium resulted in increased carotenoid production (Figure 5c), with a maximal carotenoid content of 0.020 ± 0.001 Abs_454nm_mg^−1^_DCW_ (6 g L^−1^ of carbon and 0.5 g L^−1^ of nitrogen). Temperature and pH, as well as initial sugar and nitrogen contents, had effects on the carotenoid production by the yeast *Rhodotorula mucilaginosa.* Increased growth and carotenoid concentration could be observed at higher sugar concentrations [37]. In *Rhodotorula glutinis* CCY20-2-26, the highest pigment accumulation was determined at a C:N of 20, with the content decreasing at a higher C:N ratio [74]. In contrast, the carotenoid content of *Rhodotorula glutinis* increased with an increase in the C:N ratio. Additionally, both high and low concentrations of ammonium had negative effects on carotenoid production [73].

In RSM, low experiment numbers are sufficient to find the optimum parameters, as well as interpret effects among variables. However, this method also has drawbacks [24,29]. Applied quadratic non-linear correlations might not be sufficiently complex to explain non-linear dependencies in biological processes [29]. The CCD of different carbon and nitrogen concentrations presented in this study allows an estimation of the influence on the biomass, lipid, and carotenoid production of *R. erythropolis.* Furthermore, a suitable setup, enabling both biomass and lipid production, could be identified. One possibility is to achieve high-density biomass using high carbon and nitrogen concentrations supplied to the media, followed by nitrogen limitation for lipid accumulation.

## 4. Conclusions

In this study, a threefold media optimization method was performed for *R. erythropolis* using selected carbon and nitrogen sources. The data generated in this work, together with the CCD, provide a comprehensive source of information for the identification of suitable and cost-efficient feedstocks used in industrial processes. For the first time, the effects of carbon and nitrogen concentrations on the production of lipids and carotenoids in *R. erythropolis* have been assessed via a response surface model. Comparisons between fatty acid profiles when grown on different carbon and nitrogen sources revealed the enhanced production of oleic acid and stearic acid when grown on yeast extract and tryptone/peptone. Higher amounts of odd-chain fatty acids were detected when grown on lactose or maltose. Odd-chain fatty acids have unique pharmacological functions and are positively related to human health; they are, therefore, of high interest [75]. The optimal culture composition developed here could aid future studies that aim to understand de novo lipogenesis in *R. erythropolis*. Further, asymmetric carotenoids are of great industrial interest, as they are very difficult to chemically produce, but they can be produced by a lycopene β-cyclase identified in *R. erythropolis* AN12 [8,18].

Applying a C:N of 100, nine nitrogen and eight carbon sources were investigated. As all other parameters were held constant, new insights about the effects of a wide variety of carbon and nitrogen sources could be concluded. Maximum biomass acquisition was obtained using the nitrogen source ammonium acetate, whereas the complex nitrogen source yeast extract, followed by tryptone/peptone, achieved the highest lipid yield. With respect to different carbon sources, the highest biomasses could be observed using glucose, followed by sucrose. Notably, sugar alcohols, sorbitol and glycerol induced elevated lipid titers. While biomass production increased with the increase in carbon and nitrogen concentrations in the presented CCD, lipid production decreased.

The results of this study facilitate the identification of suitable waste stream substrates for *R. erythropolis* JCM3201^T^, as insights into the substrate flexibility and nutrient flow could serve as cultivation references for future studies of this promising organism. Future work could apply more sophisticated modeling solutions, such as machine learning algorithms, i.e. Bayesian optimization, for the optimization of this strain at a larger scale. To this end, additional factors, such as optimal pH, temperature, pO_2_, vitamins, and trace elements, could also be identified in a time- and cost-efficient manner. This current work could also feed into a big data acquisition model, where RSM and other methodologies could be extracted from various works to capture the non-linear behavior of a biological system.

## Figures and Tables

**Figure 1 microorganisms-11-02147-f001:**
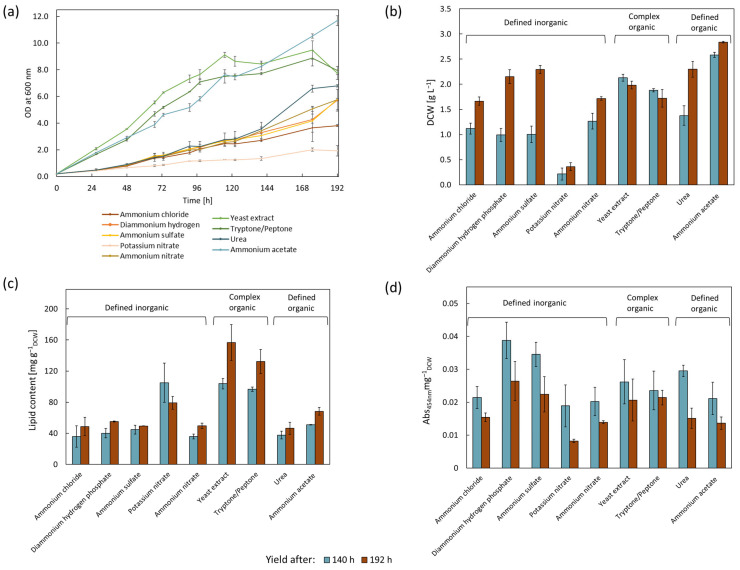
Effects of various nitrogen sources on growth curves, as well as biomass, lipid, and carotenoid accumulation of *R. erythropolis* after 140 and 192 h. Yield after 140 h is depicted in blue, and yield after 192 h is depicted in red. Samples were cultured using variable nitrogen sources at a nitrogen concentration of 0.16 g L^−1^ (C:N of 100, 16 g L^−1^ of carbon in the form of glucose), *n* = 3. (**a**) Cell growth measured as OD at 600 nm over 192 h. (**b**) Dry cell weight (DCW) at 140 and 192 h. (**c**) Lipid content (normalized to DCW) at 140 and 192 h. (**d**) Carotenoid accumulation (normalized to DCW) at 140 and 192 h.

**Figure 2 microorganisms-11-02147-f002:**
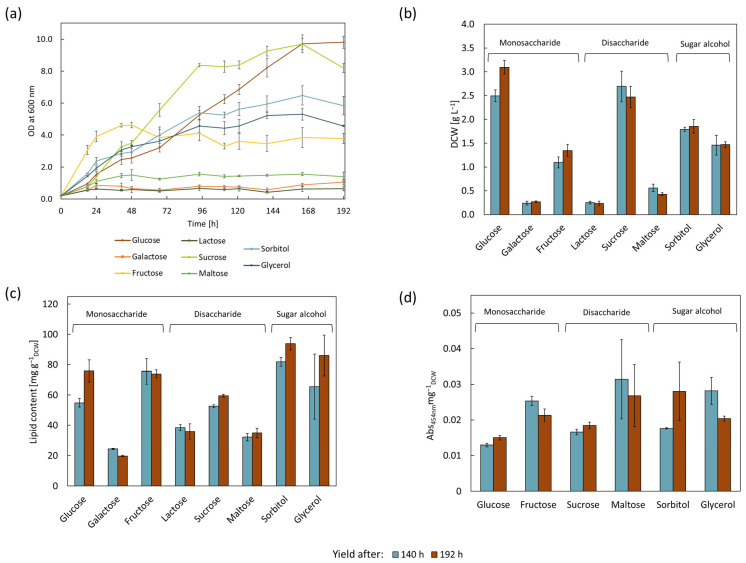
Effect of various carbon sources on growth, lipid, and carotenoid accumulation of *R. erythropolis* after 140 and 192 h of being cultured in parallel with carbon concentration of 16 g L^−1^, holding all other conditions to be identical (C:N of 100, 0.16 g L^−1^ of nitrogen in the form of ammonium acetate), *n* = 3. Yield after 140 h is depicted in blue, and yield after 192 h is depicted in red. (**a**) Growth measured as OD at 600 nm over 192 h. (**b**) Dry cell weight (DCW) at 140 and 192 h. (**c**) Lipid content (normalized to DCW) at 140 and 192 h. (**d**) Carotenoid accumulation (normalized to DCW) at 140 and 192 h. No carotenoid extraction was performed for galactose and lactose due to a lack of sufficient biomass formation.

**Figure 3 microorganisms-11-02147-f003:**
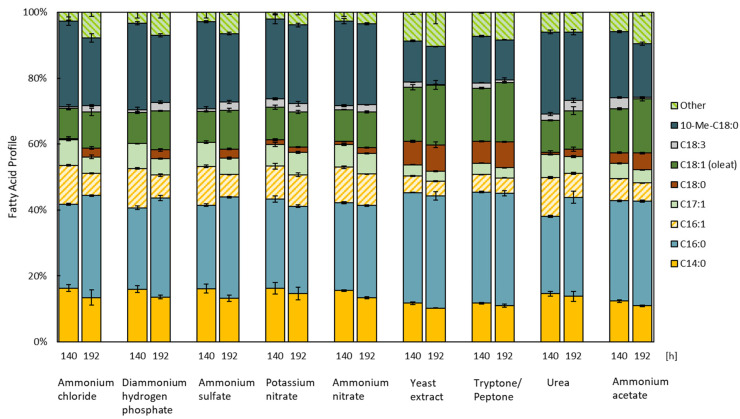
Fatty acid profiles of *R. erythropolis* grown using distinct nitrogen sources (*n* = 3). “Other” constitutes fatty acids with a total fatty acid content of representation below 3% (*w*/*w*) and includes C14:1, C15:0, C17:0, 14-Methyl-C16:0, C20:1, C20:3, C20:5, and C22:1. For each group, the two timepoints—140 h (left) and 192 h (right)—are depicted.

**Figure 4 microorganisms-11-02147-f004:**
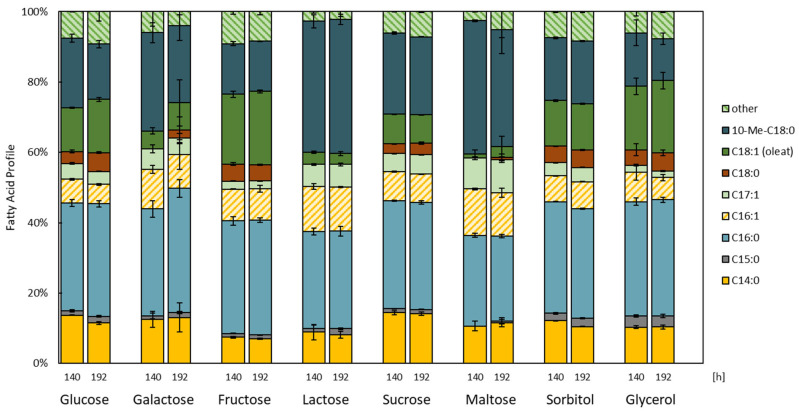
Fatty acid profiles of *R. erythropolis* grown using distinct carbon sources (*n* = 3, except for Maltose after 192 h, where *n* = 2, as one sample was excluded as an outlier). “Other” constitutes fatty acids with a total fatty acid content representation of below 3% (*w*/*w*) and includes C14:1, C17:0, 14-Methyl-C16:0, C18:3, C20:1, C20:3, C20:5, and C22:1, among others. For each group, the two timepoints—140 h (left) and 192 h (right)—are depicted.

**Figure 5 microorganisms-11-02147-f005:**
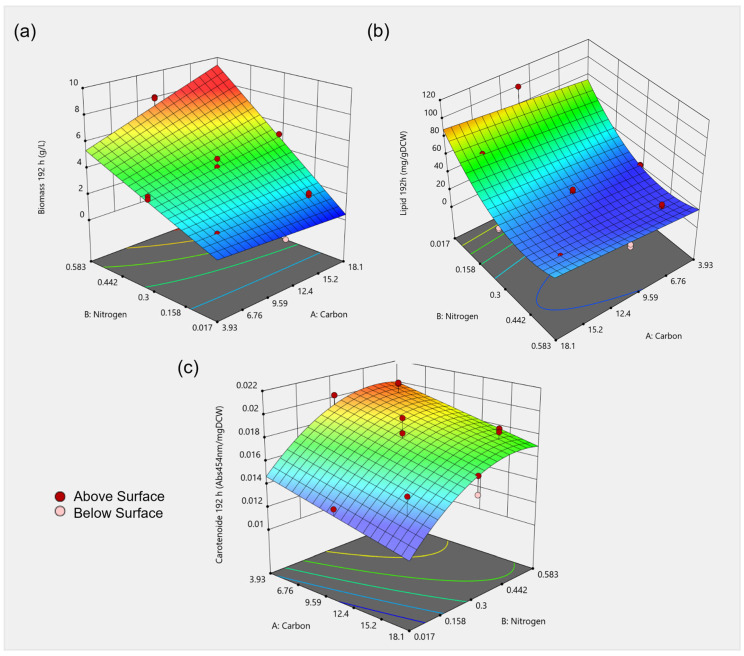
A 3D response surface plot of the combined effect of carbon and nitrogen levels on (**a**) biomass (g L^−1^), (**b**) lipid content (mg g^−1^_DCW_) and (**c**) carotenoid content (Abs_454nm_mg^−1^_DCW_) after 192 h. Red points mark the measured points on which the model was built, dark red points were measured above the depicted model surface and light red points were measured below the model surface.

**Table 1 microorganisms-11-02147-t001:** Matrix of tested nitrogen and carbon sources, as well as their chemical nature.

Number	Nitrogen Source	Chemical Nature	Carbon Source	Chemical Nature
1	Ammonium chloride	Defined inorganic	Glucose	Monosaccharides
2	Diammonium hydrogen phosphate		Galactose	
3	Ammonium sulfate		Fructose	
4	Potassium nitrate		Lactose	Disaccharides
5	Ammonium nitrate		Sucrose	
6	Yeast extract	Complex organic	Maltose	
7	Tryptone/peptone		Sorbitol	Sugar alcohol
8	Urea	Defined organic	Glycerol	
9	Ammonium acetate			

**Table 2 microorganisms-11-02147-t002:** Range of values for central composite design.

Variable	Level				
	−1.414	−1	0	+1	+1.414
Carbon (g L^−1^)	3.929	6	11	16	18.071
Nitrogen (g L^−1^)	0.017	0.1	0.3	0.5	0.583

**Table 3 microorganisms-11-02147-t003:** Full central composite design for the optimization of carbon and nitrogen concentrations in the cultivation media of *R. erythropolis*. C:N ratio, with C (carbon concentration) and N (nitrogen concentration), as well as the determined biomass, lipid, and carotenoid contents after 140 and 192 h, are listed. Each run was analyzed in two technical replicates. Due to technical complications, results of Run 13 represent one biological sample.

Run	C:N	C g L^−1^	N g L^−1^	Biomass g L^−1^	Lipid Content mg g^−1^_DCW_	Carotenoid Content Abs_454nm_mg^−1^_DCW_	Biomass g L^−1^	Lipid Content mg g^−1^_DCW_	Carotenoid Content Abs_454nm_mg^−1^_DCW_
				140 h			192 h		
1	60.00	6.000	0.100	1.44	26.90	0.0192	1.72	43.89	0.0159
2	60.00	6.000	0.100	1.23	25.86	0.0175	1.78	40.79	0.0148
3	160.00	16.000	0.100	1.74	41.83	0.0156	1.94	63.19	0.0156
4	160.00	16.000	0.100	1.73	40.09	0.0146	2.12	65.65	0.0133
5	12.00	6.000	0.500	3.21	30.61	0.0124	4.82	22.34	0.0206
6	12.00	6.000	0.500	3.48	30.26	0.0146	4.60	24.36	0.0207
7	32.00	16.000	0.500	4.02	28.18	0.0165	7.43	28.09	0.0186
8	32.00	16.000	0.500	5.12	27.60	0.0156	7.25	25.14	0.0183
9	13.10	3.929	0.300	2.49	27.11	0.0134	3.66	22.01	0.0186
10	13.10	3.929	0.300	2.33	26.76	0.0154	3.45	21.34	0.0202
11	60.24	18.071	0.300	3.19	28.53	0.0238	4.53	29.71	0.0164
12	60.24	18.071	0.300	3.61	26.63	0.0209	5.13	31.30	0.0148
13	647.06	11.000	0.017	0.63	65.72	0.0132	0.50	100.53	0.0138
14	18.87	11.000	0.583	5.37	30.13	0.0173	8.04	22.60	0.0172
15	18.87	11.000	0.583	5.50	27.77	0.0183	7.96	25.68	0.0179
16	36.67	11.000	0.300	2.73	26.96	0.0190	3.96	28.79	0.0196
17	36.67	11.000	0.300	2.83	27.42	0.0173	4.76	29.52	0.0172
18	36.67	11.000	0.300	2.80	26.18	0.0184	4.12	32.76	0.0170
19	36.67	11.000	0.300	2.71	25.23	0.0166	4.05	31.79	0.0183
20	36.67	11.000	0.300	2.77	26.80	0.0189	3.20	33.16	0.0174

**Table 4 microorganisms-11-02147-t004:** ANOVA of full central composite design (CCD) models for the optimization of carbon and nitrogen concentrations in *R. erythropolis* cultivation media.

Response		Model	*p*-Value Model	*p*-Value Lack of Fit	R^2^	Adjusted R^2^	Predicted R^2^
Biomass	140 h	Linear	<0.0001	0.0318	0.9306	0.9224	0.8976
	192 h	2FI	<0.0001	0.0722	0.9611	0.9538	0.9416
Lipid	140 h	Reduced Quadratic	0.0007	<0.0001	0.7058	0.6273	0.2220
	192 h	Reduced Quadratic	<0.0001	<0.0001	0.8725	0.8486	0.7262
Carotenoids	140 h	2FI	0.1085	0.0004	0.3080	0.1782	−0.1744
	192 h	Reduced Quadratic	<0.0001	0.2525	0.7936	0.7549	0.6715

## Data Availability

Not applicable.

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
