# Peer review of "Optimization of Rhodococcus erythropolis JCM3201T Nutrient Media to Improve Biomass, Lipid, and Carotenoid Yield Using Response Surface Methodology"

_microorganisms, 2023, doi:10.3390/microorganisms11092147_

Round 1

Reviewer 1 Report (Previous Reviewer 2)

Some minor technical corrections are required 

No items to correct, expect for typing

Author Response

Reviewer 2 Report (New Reviewer)

The manuscript presented by the authors shows a response surface analysis for optimizing a culture medium for Rhodococcus erythropolis to improve the production of biomass, lipids, and carotenoids. It is an exciting topic. However, there are several aspects to improve in the manuscript for publication; the document comes with comments and corrections by the author, which make the manuscript complex to read and evaluate.

1. In the title, the phrase statistically based can be omitted, as implementing a response surface analysis implies the use of statistical knowledge.

2. In the abstract, it is essential to indicate which type of response surface analysis was used, which variables were analyzed, and which were the response variables. Finally, it is necessary to show the total number of experiments and how many replicates were performed.

3. In the introduction, the sentence in line 95 is repeated in line 98; it is suggested to leave only one sentence.

4. In the introduction, it is necessary to give a context that other types of microorganisms can produce carotenoids and lipids, for example, microalgae and cyanobacteria among different organisms, and indicate the advantages and disadvantages of Rhodococcus erythropolis.

5. In the introduction, the authors must clearly show the work's innovative contribution and what aspects it addresses that have not yet been studied.

6. In the methodology, line 158 should eliminate the doubled.

7. In the experimental design section, it is suggested that the authors make a table showing the evaluated variables: the lower limit, the upper limit, and the central value for each variable; this allows a better understanding of the design.

8. In section 2.4, it is important to indicate how often the samples were taken to determine the biomass concentration.

9. The Pigment Extraction section suggests that the authors include the formula for calculating measured carotenoids. It is recommended that the section's title be changed to Carotenoid Extraction.

10. Figure 1 is very confusing; the legend does not clearly show what the authors want to indicate. In graphs b, c, and d, the two bars for each nitrogen source mean initial and final concentration.

11. It is necessary to expand the discussion of the effect of the nitrogen source on carotenoid production; it is clear that nutrient stress, mainly of nitrogen and phosphorus influences the concentration of carotenoids; the authors must try to explain why some nitrogen sources favor the accumulation of carotenoids.

12. It is necessary that the authors submit the final version of the manuscript without comments so that it can be better read and evaluated.

13. Since it is an optimization, the surface plots for biomass and lipids are not yet optimized. This shows the need for a new design with the values expressed in the software optimization. Further experimentation is suggested to determine the optimal concentrations concretely. The program allows maximizing the production of one or all of the response variables; it would be interesting if the authors could provide a table with the optimal concentrations that optimize production and that these could be validated experimentally.

Round 2

Reviewer 2 Report (New Reviewer)

The manuscript submitted by the authors integrated the comments made in the previous version. It is suggested that the manuscript be published. However, table 2 should be stated in the experimental design section, and Figure 3 should be enlarged. 

This manuscript is a resubmission of an earlier submission. The following is a list of the peer review reports and author responses from that submission.

Round 1

Reviewer 1 Report

In this study, the authors used the carotenogenic and oleaginous bacterium Rhodococcus erythropolis JCM3201 and employed response surface methodology (RSM) to determine optimal conditions to produce lipid and carotenoid by this bacterium. RSM is a powerful tool to determine optimal conditions for microbial growth and/or production of useful compounds and has also been used generally in the field of microbiology. So, I think that the specific targets are important for the research using RSM.

As the authors mentioned, Rhodococcus sp. bacteria can be possible effective platforms in bioconversion and bioremediation, and RSM has also been adopted in the application of Rhodococcus sp. bacteria as follows:

Nitrile hydratase, Heliyon 6:e05111, 2020; Arsenite bioremediation, Sci Total Environ 687:577-589, 2019; PHB production, Sci Rep 11:1896, 2021…

Thus, RSM should be used for microbial production or bioremediation of the specific compounds. What kinds of lipid and carotenoid did the authors analyze in this study? While increasing biomass is one of the purposes in this study, required state of biomass should be different in each application. I would like to know the novelty and feasibility of this study.

Reviewer 2 Report

This is an interesting and well-designed study that demonstrates dependence of lipid and pigment synthesis by biotechnologically important bacterium Rhodococcus erythropolis upon carbon-to-nitrogen ratios in the medium, as well as C and N sources. The major drawback is that the manuscript in its current form is too descriptive and lengthy written, especially in the section for Results. The authors should substantially overwork and edit their manuscript in view of numerous points to be corrected and problems with terminology.

1.     The authors should highlight more the novelty and significance of this study than in the original version.

2.     It is unclear from the ms what novel result was produced by using RSM and CCD in this work. Is this improved experimental design without necessity to include numerous factors? On the other hand, it seems that RSM and CCD were used as good statistical tool as already discussed (Bertrand B et al. Statistical Design, a Powerful Tool for Optimizing Biosurfactant Production: A Review. Colloids and Interfaces. 2018; 2(3):36).

3.     The authors may consider another title, such as “Optimization of Rhodococcus erythropolis JCM3201 nutrient media to improve biomass, lipid, and carotenoid yield using statistics-based response surface methodology

4.     RSM should be spelled out in the original title.

5.     The concluding sentence in Abstract “The presented results provide new insights into the physiology of Rhodococcus erythropolis under variable nutritional states” is too general and should be focused on the usefulness for biotechnological applications.

6.     Subsection for RSM and CCD (L 72-79) should be written more clearly than in the original text or even moved to the description of the experimental design. Does “the number of runs” mean here a set of experimental conditions to be studied? Are RSM and CCD innovative or commonly used methods in studies of rhodococci?

7.     Rhodococcus erythropolis JCM3201 is type strain, and superscript T should added to the strain designation (L89) as JCM3201T

8.     Overall, description of results is too long and should certainly be reduced.

9.     .I recommend the authors to focus on general comparisons of the bacterial growth in media with different carbon and nitrogen sources/ratios and to provide some (not all) numerical data for the most important (contrast) cases.

10.  Discussion should be in the separate section, not be mixed with results.

11.  Conclusions should be concentrated on the significance and prospective of this study, not comparisons with the reported data. The novelty is still uclear.

The last sentence “In the future, the strain can additionally be developed by random mutagenesis [69] or genetic engineering” is not related to the results of this study.

12.  The authors have not explained why two time points, 140 h and 192 h, have been chosen and always compared. Time points that corresponded to the stationary phase onset differed depending on the nutritional source used. Is this necessary to compare the data for these points everywhere?

13.  The authors should carefully check whether cells continued to divide and grow from 140 h to 192 h. Many bacterial cultures entered the stationary phase earlier than by 140 h and growth no longer occurred. Some cells might undergo lysis; it is no wonder that DCW was declining over prolonged incubation in post-stationary cultures. It is useful to indicate the onset of stationary phase with arrows in plots and to compare in the text.

14.  Let’s consider an example: “After 140 h, a DCW of around one gram per litre could be detected for ammonium chloride (1.12 ± 0.11 g L-1), diammoniumhydrogenphosphat (0.99 ± 0.13 g L-1) and ammonium sulfate (1 ± 0.17 g L-1). Cultures supplemented with ammonium nitrate showed a higher DCW of 1.27 ± 0.16 g L-1. The lowest biomass accumulation was recorded for potassium nitrate as nitrogen source, with an OD600 of 1.35 and a DCW of 0.22 ± 0.12 g L-1 after 140 h. After 192 h, a DCW of only 0.36 ± 0.08 g L-1 was reached”. My comments for this example. All this information can be seen from plots; the text should not describe all details from illustrations. DCW was determined, not “could be detected”. As for “culture supplemented”, please note that supplementation of a culture with a substance means commonly that this substance is added to a growing or already grown culture. Here, the starting medium contained this N source, and the starting medium and the culture that was grown in it are not the same. The verb recorded should be replaced with achieved.

15.  Similar and other rough phrases and formulations should carefully be edited through the text and captions. Below are many points (not all, of course) that need corrections.

16.  Plots 1b and 2b should be divided in two plots: they are difficult to read.

17.  The authors often write elemental carbon or nitrogen; “elemental” can be omitted.

18.  Please replace “seed cultures” for inoculum,

19.  A term “triggering” should be used for a biochemical process; not for accumulation.

20.  Please rephrase “growth is impaired”

21.  media was -> media were

22.  Please use the Past Tense for all verbs in Methods

23.  All cultivations were performed in -> All cultures were cultivated in

24.  Please edit “C:N of elemental carbon to nitrogen weight were covered”

25.  It is enough to use storage compounds [1], instead storage compounds for carbon and energy [1].

26.  Check spelling for diammoniumhydrogenphosphat à diammonium hydrogen phosphate through the text

27.  Please rephrase: For each nitrogen and carbon source, biological triplicates were inoculated…

28.  Please explain or make clear the sentence “A total of 21 runs with different carbon-to-nitrogen ratios were prepared in a CCD based …”

29.  Please edit alle other ratios > all other ratios. Explain “were conducted”. This sentence is unclear.

30.  The sentence “Each run was analysed in two technical replicates each” is unclear.

31.  Optical density was determined -> Optical density was measured. Path length of cuvette should be indicated instead its volume.

32.  Please spell out DCW and DOE,

33.  Please correct “ with the modification of larger glass beads (2 mm)”.

34.  Please rephrase “for design analysis of the media optimization of C:N”.

35.  Please correct “experiments investigating different nitrogen and carbon sources”. An experiment cannot investigate.

36.  It is too obvious that “Although ammonium acetate is an organic compound, the nitrogen source ammonium is inorganic”.I recommend avoiding such banal information. It holds true for all the text.

37.  Nonsence: The five defined inorganic sources exhibited slow growth at the beginning. An inorganic source cannot exhibit growth, this is a feature of bacterial culture.

38.  Biomass formation -> Yield of biomass

39.  Legends to Fig. 1 and Fig. 2 look too similar and should be rewritten so to make clearer the difference between these experimental sets. For the first legend: glucose (as the same carbon source) and various N sources, indicate then  ratios and concentrations). For the second legend: various C sources and the same N-source (ammonium acetate), indicate then  then then ratios and concentrations

I recommed the authors to ask the expert in bacterial physiology to help in editing the text and terminology. Indded, I found numerous weak points that should be corrected.